# Exploiting Argument Information to Improve Event Detection via Supervised Attention Mechanisms

## Abstract

This paper tackles the task of event detection (ED), which involves identifying and categorizing events. We argue that arguments provide significant clues to this task, but they are either completely ignored or exploited in an indirect manner in existing detection approaches. In this work, we propose to exploit argument information explicitly for ED via supervised attention mechanisms. In specific, we systematically investigate the proposed model under the supervision of different attention strategies. Experimental results show that our approach advances state-of-the-arts and achieves the best $F_1$ score on ACE 2005 dataset.

## 1 Introduction

In the ACE (Automatic Context Extraction) event extraction program, an event is represented as a structure comprising an **event trigger** and a set of **arguments**. This work tackles event detection (ED) task, which is a crucial part of event extraction (EE) and focuses on identifying event triggers and categorizing them. For instance, in the sentence "**He *died* in the hospital**", an ED system is expected to detect a ***Die*** event along with the trigger word "*died*". Besides, the task of EE also includes event argument extraction (AE), which involves event argument identification and role classification. In the above sentence, the arguments of the event include "He"($Role = Person$) and "hospital"($Role = Place$). However, this paper does not focus on AE and only tackles the former task.

According to the above definitions, event arguments seem to be not essentially necessary to ED. However, we argue that they are capable of providing significant clues for identifying and categorizing events. They are especially useful for ambiguous trigger words. For example, consider a sentence in ACE 2005 dataset:

> **Mohamad *fired* Anwar**, his **former protege**, in **1998**.

In this sentence, "*fired*" is the trigger word and the other bold words are event arguments. The correct type of the event triggered by "*fired*" in this case is ***End-Position***. However, it might be easily misidentified as ***Attack*** because "*fired*" is a multivocal word. In this case, if we consider the phrase "former protege", which serves as an argument ($Role = Position$) of the target event, we would have more confidence in predicting it as an ***End-Position*** event.

Unfortunately, most exiting methods performed event detection individually, where the annotated arguments in training set are totally ignored (Ji and Grishman, 2008; Gupta and Ji, 2009; Hong et al., 2011; Chen et al., 2015; Nguyen and Grishman, 2015; Liu et al., 2016a,b; Nguyen and Grishman, 2016). Although some joint learning based methods have been proposed, which tackled event detection and argument extraction simultaneously (Riedel et al., 2009; Li et al., 2013; Venugopal et al., 2014; Nguyen et al., 2016), these approaches usually only make remarkable improvements to AE, but insignificant to ED. Table 1 illustrates our observations. Li et al. (2013) and Nguyen et al. (2016) are state-of-the-art joint models in symbolic and embedding methods for event extraction, respectively. Compared with state-of-the-art pipeline systems, both join-

| Methods | | ED | AE |
|---|---|---|---|
| Symbolic | Hong's pipeline (2011) | 68.3 | 48.3 |
| Methods | Li's joint (2013) | 67.5 | 52.7 |
| Embedding | Chen's pipeline (2015) | 69.1 | 53.5 |
| Methods | Nguyen's joint (2016) | 69.3 | 55.4 |

Table 1: Performances of pipeline and joint approaches on ACE 2005 dataset. The pipeline method in each group was the state-of-the-art system when the corresponding joint method was proposed.

t methods achieved remarkable improvements on AE (over 1.9 points), whereas achieved insignificant improvements on ED (less than 0.2 points). The symbolic joint method even performed worse (67.5 vs. 68.3) than pipeline system on ED.

We believe that this phenomenon may be caused by the following two reasons. On the one hand, since joint methods simultaneously solve ED and AE, methods following this paradigm usually combine the loss functions of these two tasks and are jointly trained under the supervision of annotated triggers and arguments. However, training corpus contains much more annotated arguments than triggers (about 9800 arguments and 5300 triggers in ACE 2005 dataset) because each trigger may be along with multiple event arguments. Thus, the unbalanced data may cause joint models to favor AE task. On the other hand, in implementation, joint models usually pre-predict several potential triggers and arguments first and then make global inference to select correct items. When pre-predicting potential triggers, almost all existing approaches do not leverage any argument information. In this way, ED does hardly benefit from the annotated arguments. By contrast, the component for pre-prediction of arguments always exploits the extracted trigger information. Thus, we argue that annotated arguments are actually used for AE, not for ED in existing joint methods, which is also the reason we call it an indirect way to use arguments for ED.

Contrast to joint methods, this paper proposes to exploit argument information explicitly for ED. We have analyzed that arguments are capable of providing significant clues to ED, which gives us an enlightenment that ar-

guments should be focused on when performing this task. Therefore, we propose a neural network based approach to detect events in texts. And in the proposed approach, we adopt a supervised attention mechanism to achieve this goal, where argument words are expected to acquire more attention than other words. The attention value of each word in a given sentence is calculated by an operation between the current word and the target trigger candidate. Specifically, in training procedure, we first construct gold attentions for each trigger candidate based on annotated arguments. Then, treating gold attentions as the supervision to train the attention mechanism, we learn attention and event detector jointly both in supervised manner. In testing procedure, we use the ED model with learned attention mechanisms to detect events.

In the experiment section, we systematically conduct comparisons on a widely used benchmark dataset ACE2005[1]. In order to further demonstrate the effectiveness of our approach, we also use events from FrameNet (FN) (F. Baker et al., 1998) as extra training data, as the same as Liu et al. (2016a) to alleviate the data-sparseness problem for ED to augment the performance of the proposed approach. The experimental results demonstrate that the proposed approach is effective for ED task, and it outperforms state-of-the-art approaches with remarkable gains.

To sum up, our main contributions are: (1) we analyze the problem of joint models on the task of ED, and propose to use the annotated argument information explicitly for this task. (2) to achieve this goal, we introduce a supervised attention based ED model. Furthermore, we systematically investigate different attention strategies for the proposed model. (3) we improve the performance of ED and achieve the best performance on the widely used benchmark dataset ACE 2005.

## 2 Task Description

The ED task is a subtask of ACE event evaluations where an event is defined as a specific occurrence involving one or more participants. Event extraction task requires certain specified types of events, which are mentioned

---

[1]https://catalog.ldc.upenn.edu/LDC2006T06

in the source language data, be detected. We firstly introduce some ACE terminologies to facilitate the understanding of this task:

**Entity**: an object or a set of objects in one of the semantic categories of interests.

**Entity mention**: a reference to an entity (typically, a noun phrase).

**Event trigger**: the main word that most clearly expresses an event occurrence.

**Event arguments**: the mentions that are involved in an event (participants).

**Event mention**: a phrase or sentence within which an event is described, including the trigger and arguments.

The goal of ED is to identify event triggers and categorize their event types. For instance, in the sentence "**He died** in the **hospital**", an ED system is expected to detect a ***Die*** event along with the trigger word "*died*". The detection of event arguments "He"($Role = Person$) and "hospital"($Role = Place$) is not involved in the ED task. The 2005 ACE evaluation included 8 super types of events, with 33 sub-types. Following previous work, we treat these simply as 33 separate event types and ignore the hierarchical structure among them.

## 3 The Proposed Approach

Similar to existing work, we model ED as a multi-class classification task. In detail, given a sentence, we treat every token in that sentence as a trigger candidate, and our goal is to classify each of these candidates into one of 34 classes (33 event types plus an NA class).

In our approach, every word along with its context, which includes the contextual words and entities, constitute an event trigger candidate. Figure 1 describes the architecture of the proposed approach, which involves two components: (i) Context Representation Learning (CRL), which reveals the representation of both contextual words and entities via attention mechanisms; (ii) Event Detector (ED), which assigns an event type (including the NA type) to each candidate based on the learned contextual representations.

### 3.1 Context Representation Learning

In order to prepare for Context Representation Learning (CRL), we limit the context to a fixed length by trimming longer sen-

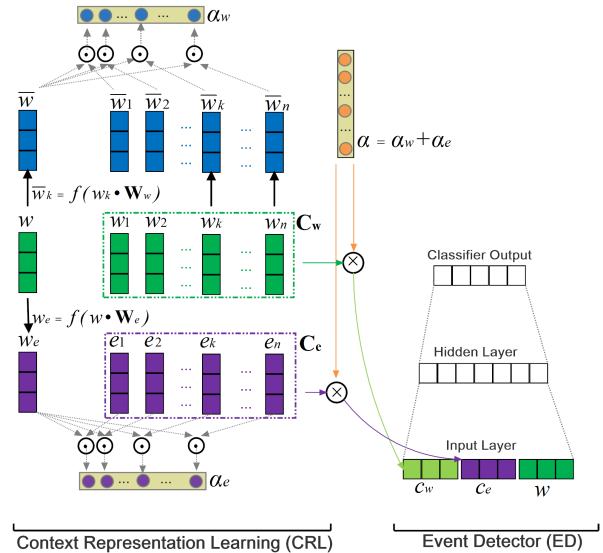

Figure 1: The architecture of the proposed approach for event detection. In this figure, $w$ is the candidate word, $[w_1, ..., w_n]$ is the contextual words of $w$, and $[e_1, ..., e_n]$ is the corresponding entity types of $[w_1, ... , w_n]$.

tences and padding shorter sentences with a special token when necessary. Let $n$ be the fixed length and $w_0$ be the current candidate trigger word, then its contextual words $\mathbf{C_w}$ is $[w_{-\frac{n}{2}}, w_{-\frac{n}{2}+1}, ..., w_{-1}, w_1, ..., w_{\frac{n}{2}-1}, w_{\frac{n}{2}}]$[2], and its contextual entities, which is the corresponding entity types (including an NA type) of $\mathbf{C_w}$, is $[e_{-\frac{n}{2}}, e_{-\frac{n}{2}+1}, ..., e_{-1}, e_1, ..., e_{\frac{n}{2}-1}, e_{\frac{n}{2}}]$. For convenience, we use $w$ to denote the current word, $[w_1, w_2, ..., w_n]$ to denote the contextual words $\mathbf{C_w}$ and $[e_1, e_2, ..., e_n]$ to denote the contextual entities $\mathbf{C_e}$ in figure 1. Note that, both $w$, $\mathbf{C_w}$ and $\mathbf{C_e}$ mentioned above are originally in symbolic representation. Before entering CRL component, we transform them into real-valued vector by looking up word embedding table and entity type embedding table. Then we calculate attention vectors for both contextual words and entities by performing operations between the current word $w$ and its contexts. Finally, the contextual words representation $c_w$ and contextual entities representation $c_e$ are formed by the weighted sum of the corresponding embeddings of each word and entity in $\mathbf{C_w}$ and $\mathbf{C_e}$, respectively. We will give the details in the fol-

---

[2]The current candidate trigger word $w_0$ is not included in the context.

lowing subsections.

### 3.1.1 Word Embedding Table

Word embeddings learned from a large amount of unlabeled data have been shown to be able to capture the meaningful semantic regularities of words (Bengio et al., 2003; Erhan et al., 2010). This paper uses the learned word embeddings as the source of basic features. Specifically, we use the Skip-gram model (Mikolov et al., 2013) to learn word embeddings on the NYT corpus[3].

### 3.1.2 Entity Type Embedding Table

The ACE 2005 corpus annotated not only events but also entities for each given sentence. Following existing work (Li et al., 2013; Chen et al., 2015; Nguyen and Grishman, 2015), we exploit the annotated entity information in our ED system. We randomly initialize embedding vector for each entity type (including the NA type) and update it in training procedure.

### 3.1.3 Representation Learning

In this subsection, we illustrate our proposed approach to learn representations of both contextual words and entities, which serve as inputs to the following event detector component. Recall that, we use the matrix $\mathbf{C_w}$ and $\mathbf{C_e}$ to denote contextual words and contextual entities, respectively.

As illustrated in figure 1, the CRL component needs three inputs: the current candidate trigger word $w$, the contextual words $\mathbf{C_w}$ and the contextual entities $\mathbf{C_e}$. Then, two attention vectors, which reflect different aspects of the context, are calculated in the next step.

**The contextual word attention vector** $\alpha_w$ is computed based on the current word $w$ and its contextual words $\mathbf{C_w}$. We firstly transform each word $w_k$ (including $w$ and every word in $\mathbf{C_w}$) into a hidden representation $\overline{w}_k$ by the following equation:

$$\overline{w}_k = f(w_k \cdot W_w) \qquad (1)$$

where $f(\cdot)$ is a non-linear function such as the hyperbolic tangent, and $W_w$ is the transformation matrix. Then, we use the hidden representations to compute the attention value for each

---

[3]https://catalog.ldc.upenn.edu/LDC2008T19

word in $\mathbf{C_w}$:

$$\alpha_w^k = \frac{\exp(\overline{w} \cdot \overline{w}_k^{\mathrm{T}})}{\sum_i \exp(\overline{w} \cdot \overline{w}_i^{\mathrm{T}})} \qquad (2)$$

**The contextual entity attention vector** $\alpha_e$ is calculated with a similar method to $\alpha_w$.

$$\alpha_e^k = \frac{\exp(w_e \cdot e_k^{\mathrm{T}})}{\sum_i \exp(w_e \cdot e_i^{\mathrm{T}})} \qquad (3)$$

Note that, we do not use the entity information of the current candidate token to compute the attention vector. The reason is that only a small percentage of true event triggers are entities[4]. Therefore, the entity type of a candidate trigger is meaningless for ED. Instead, we use $w_e$, which is calculated by transforming $w$ from the word space into the entity type space, as the attention source.

We combine $\alpha_w$ and $\alpha_e$ to obtain the final attention vector, $\alpha = \alpha_w + \alpha_e$. Finally, the contextual words representation $c_w$ and the contextual entities representation $c_e$ are formed by weighted sum of $\mathbf{C_w}$ and $\mathbf{C_e}$, respectively:

$$c_w = \mathbf{C_w}\alpha^{\mathrm{T}} \qquad (4)$$

$$c_e = \mathbf{C_e}\alpha^{\mathrm{T}} \qquad (5)$$

## 3.2 Event Detector

As illustrated in figure 1, we employ a three-layer (an input layer, a hidden layer and a softmax output layer) Artificial Neural Networks (ANNs) (Hagan et al., 1996) to model the ED task, which has been demonstrated very effective for event detection by Liu et al. (2016a).

### 3.2.1 Basic ED Model

Given a sentence, as illustrated in figure 1, we concatenate the embedding vectors of the context (including contextual words and entities) and the current candidate trigger to serve as the input to ED model. Then, for a given input sample $\mathbf{x}$, ANN with parameter $\theta$ outputs a vector $\mathbf{O}$, where the $i$-th value $o_i$ of $\mathbf{O}$ is the confident score for classifying $\mathbf{x}$ to the $i$-th event type. To obtain the conditional probability $p(i|\mathbf{x}, \theta)$, we apply a softmax operation over all event types:

$$p(i|\mathbf{x}, \theta) = \frac{e^{o_i}}{\sum_{k=1}^m e^{o_k}} \qquad (6)$$

---

[4]Only 10% of triggers in ACE 2005 are entities.

Given all of our (suppose T) training instances $(\mathbf{x^{(i)}}; y^{(i)})$, we can then define the negative log-likelihood loss function:

$$J(\theta) = -\sum_{i=1}^{T} \log p(y^{(i)}|\mathbf{x^{(i)}}, \theta) \qquad (7)$$

We train the model by using a simple optimization technique called stochastic gradient descent (SGD) over shuffled mini-batches with the Adadelta rule (Zeiler, 2012). Regularization is implemented by a dropout (Kim, 2014; Hinton et al., 2012) and $L_2$ norm.

### 3.2.2 Supervised Attention

In this subsection, we introduce supervised attention to explicitly use annotated argument information to improve ED. Our basic idea is simple: argument words should acquire more attention than other words. To achieve this goal, we first construct vectors using annotated arguments as the gold attentions. Then, we employ them as supervision to train the attention mechanism.

**Constructing Gold Attention Vectors**

Our goal is to encourage argument words to obtain more attention than other words. To achieve this goal, we propose two strategies to construct gold attention vectors:

**S1: only pay attention to argument words.** That is, all argument words in the given context obtain the same attention, whereas other words get no attention. For candidates without any annotated arguments in context, we force all entities to average the whole attention. Figure 2 illustrates the details, where $\alpha^*$ is the final gold attention vector.

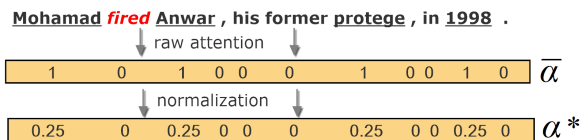

Figure 2: An example of S1 to construct gold attention vector. The word *fired* is the trigger candidate, and underline words are arguments of *fired* annotated in the corpus.

**S2: pay attention to both arguments and the words around them.** The assumption is that, not only arguments are important

to ED, the words around them are also helpful. And the nearer a word is to arguments, the more attention it should obtain. Inspired by Mi et al. (2016), we use a gaussian distribution $g(\cdot)$ to model the attention distribution of words around arguments. In detail, given an instance, we first obtain the raw attention vector $\overline{\alpha}$ in the same manner as S1 (see figure 2). Then, we create a new vector $\alpha^{'}$ with all points initialized with zero, and for each $\overline{\alpha}_i = 1$, we update $\alpha^{'}$ by adding $g(d, \mu, \sigma)$ with window size $w$, where $d$ is the distance from a word to the argument (positioned at $i$) and $\mu, \sigma$ are hyper-parameters of the gaussian distribution. Finally, similar to S1, we normalize $\alpha^{'}$ to obtain the target attention vector $\alpha^*$.

**Jointly Training ED and Attention**

Given the gold attention $\alpha^*$ (see subsection 3.2.2) and the machine attention $\alpha$ produced by our model (see subsection 3.1.3), we employ the square error as the loss function of attentions:

$$D(\theta) = \sum_{i=1}^{T} \sum_{j=1}^{n} (\alpha^{*i}_j - \alpha^i_j)^2 \qquad (8)$$

Combining equation 7 and equation 8, we define the joint loss function of our proposed model as follows:

$$J^{'}(\theta) = J(\theta) + \lambda D(\theta) \qquad (9)$$

where $\lambda$ is a hyper-parameter for trade-off between $J$ and $D$. Similar to basic ED model, we minimize the loss function $J^{'}(\theta)$ by using SGD over shuffled mini-batches with the Adadelta update rule.

## 4 Experiments

### 4.1 Dataset and Experimental Setup

**Dataset**

We conducted experiments on ACE 2005 dataset. For the purpose of comparison, we followed the evaluation of (Li et al., 2013; Chen et al., 2015; Liu et al., 2016b): randomly selected 30 articles from different genres as the development set, and subsequently conducted a blind test on a separate set of 40 ACE 2005 newswire documents. We used the remaining 529 articles as our training set.

**Hyper-parameter Setting**

Hyper-parameters are tuned on the development dataset. We set the dimension of word embeddings to 200, the dimension of entity type embeddings to 50, the size of hidden layer to 300, the output size of word transformation matrix $W_w$ in equation 1 to 200, the batch size to 100, the hyper-parameter for the $L_2$ norm to $10^{-6}$ and the dropout rate to 0.6. In addition, we use the standard normal distribution to model attention distributions of words around arguments, which means that $\mu = 0.0, \sigma = 1.0$, and the window size is set to 3 (see Subsection 3.2.2). The hyper-parameter $\lambda$ in equation 9 is various for different attention strategies, we will give its setting in the next section.

### 4.2 Correctness of Our Assumption

In this section, we conduct experiments on ACE 2005 corpus to demonstrate the correctness of our assumption that argument information is crucial to ED. To achieve this goal, we design a series of systems for comparison.

**ANN** is the basic event detection model, in which the hyper-parameter $\lambda$ is set to 0. This system does not employ argument information and computes attentions without supervision (see Subsection 3.1.3).

**ANN-ENT** assigns $\lambda$ with 0, too. The difference is that it constructs the attention vector $\alpha$ by forcing all entities in the context to average the attention instead of computing it in the manner introduced in Subsection 3.1.3. Since all arguments are entities, this system is designed to investigate the effects of entities.

**ANN-Gold1** uses the gold attentions constructed by strategy S1 in both **training** and **testing** procedure.

**ANN-Gold2** is akin to *ANN-Gold1*, but uses the second strategy to construct its gold attentions.

Note that, in order to avoid the interference of attention mechanisms, the last two systems are designed to use argument information (via gold attentions) in both training and testing procedure.

Table 2 compares these systems on ACE 2005 corpus. From the table, we observe that systems with argument information (the last two systems) significantly outperform system-

| Methods | $P$ | $R$ | $F_1$ |
|---|---|---|---|
| ANN | 73.2 | 57.9 | 64.6 |
| ANN-ENT | 79.4 | 60.7 | 68.8 |
| ANN-Gold1† | **81.9** | 65.1 | 72.5 |
| ANN-Gold2† | 81.4 | **66.9** | **73.4** |

Table 2: Experimental results on ACE 2005 corpus. † designates the systems that employ argument information.

s without argument information (the first two systems), which demonstrates that argument information is very useful for this task. Moreover, since all arguments are entities, for preciseness we also investigate that whether *ANN-Gold1/2* on earth benefits from entities or arguments. Compared with *ANN-ENT* (revising that this system only uses entity information), *ANN-Gold1/2* performs much better, which illustrates that entity information is not enough and further demonstrates that argument information is necessary for ED.

### 4.3 Results on ACE 2005 Corpus

In this section, we conduct experiments on ACE 2005 corpus to demonstrate the effectiveness of the proposed approach. Firstly, we introduce systems implemented in this work.

**ANN-S1** uses gold attentions constructed by strategy **S1** as supervision to learn attention. In our experiments, $\lambda$ is set to 1.0.

**ANN-S2** is akin to *ANN-S1*, but use strategy **S2** to construct gold attentions and the hyper-parameter $\lambda$ is set to 5.0.

These two systems both employ supervised attention mechanisms. For comparison, we use an unsupervised-attention system *ANN* as our baseline, which is introduced in Subsection 4.2. In addition, we select the following state-of-the-art methods for comparison.

1). *Li's joint model* (Li et al., 2013) extracts events based on structure prediction. It is the best structure-based system.

2). *Liu's PSL* (Liu et al., 2016b) employs both latent local and global information for event detection. It is the best-reported feature-based system.

3). *Liu's FN-Based approach* (Liu et al., 2016a) leverages the annotated corpus of FrameNet to alleviate data sparseness problem of ED based on the observation that frames in

| Methods | $P$ | $R$ | $F_1$ |
|---|---|---|---|
| Li's joint model (2013) | 73.7 | 62.3 | 67.5 |
| Liu's PSL (2016) | 75.3 | 64.4 | 69.4 |
| Liu's FN-Based (2016) | 77.6 | 65.2 | 70.7 |
| Ngyuen's joint (2016) | 66.0 | **73.0** | 69.3 |
| Skip-CNN (2016) | N/A | | 71.3 |
| ANN | 73.2 | 57.9 | 64.6 |
| ANN-S1† | **81.4** | 62.4 | 70.8 |
| ANN-S2† | 78.0 | 66.3 | **71.7** |

Table 3: Experimental results on ACE 2005. The first group illustrates the performances of state-of-the-art approaches. The second group illustrates the performances of the proposed approach. † designates the systems that employ arguments information.

FN are analogous to events in ACE.

4). *Ngyen's joint model* (Nguyen et al., 2016) employs a bi-directional RNN to jointly extract event triggers and arguments. It is the best-reported representation-based joint approach proposed on this task.

5). *Skip-CNN* (Nguyen and Grishman, 2016) introduces the non-consecutive convolution to capture non-consecutive $k$-grams for event detection. It is the best reported representation-based approach on this task.

Table 3 presents the experimental results on ACE 2005 corpus. From the table, we make the following observations:

1). *ANN* performs unexpectedly poorly, which indicates that unsupervised-attention mechanisms do not work well for ED. We believe the reason is that the training data of ACE 2005 corpus is insufficient to train a precise attention in an unsupervised manner, considering that data sparseness is an important issue of ED (Zhu et al., 2014; Liu et al., 2016a).

2). With argument information employed via supervised attention mechanisms, both *ANN-S1* and *ANN-S2* outperform *ANN* with remarkable gains, which illustrates the effectiveness of the proposed approach.

3). *ANN-S2* outperforms *ANN-S1*, but the latter achieves higher precision. It is not difficult to understand. On the one hand, strategy **S1** only focuses on argument words, which provides accurate information to identify event type, thus *ANN-S1* could achieve higher precision. On the other hand, **S2** focus-

es on both arguments and words around them, which provides more general but noised clues. Thus, *ANN-S2* achieves higher recall with a little loss of precision.

4). Compared with state-of-the-art approaches, our method *ANN-S2* achieves the best performance. We also perform a t-test ($p \leqslant 0.05$), which indicates that our method significantly outperforms all of the compared methods. Furthermore, another noticeable advantage of our approach is that it achieves much higher precision than state-of-the-arts.

## 4.4 Augmentation with FrameNet

Recently, Liu et al. (2016a) used events automatically detected from FN as extra training data to alleviate the data-sparseness problem for event detection. To further demonstrate the effectiveness of the proposed approach, we also use the events from FN to augment the performance of our approach.

In this work, we use the events published by Liu et al. (2016a)[5] as extra training data. However, their data can not be used in the proposed approach without further processing, because it lacks of both argument and entity information. Figure 3 shows several examples of this data.

##001    Grandad was [dead] by then and so were the two great-uncles .
*Event Type: Die*

##002    I [divorced] my wife for smoking in the toilet !
*Event Type: Divorce*

##003    He was [executed] yesterday.
*Event Type: Execute*

Figure 3: Examples of events detected from FrameNet (published by Liu et al. (2016a)).

**Processing of Events from FN**

Liu et al. (2016a) detected events from FrameNet based on the observation that frames in FN are analogous to events in ACE (lexical unit of a frame $\leftrightarrow$ trigger of an event, frame elements of a frame $\leftrightarrow$ arguments of an event). All events they published are also frames in FN. Thus, we treat frame elements annotated in FN corpus as event arguments. Since frames generally contain more frame elements than events, we only use core[6] elements

---

[5]https://github.com/subacl/acl16

[6]FrameNet classifies frame elements into three groups: core, peripheral and extra-thematic.

in this work. Moreover, to obtain entity information, we use RPI Joint Information Extraction System[7] (Li et al., 2013, 2014; Li and Ji, 2014) to label ACE entity mentions.

**Experimental Results**

We use the events from FN as extra training data and keep the development and test datasets unchanged.Table 4 presents the experimental results.

| Methods | $P$ | $R$ | $F_1$ |
|---|---|---|---|
| ANN | 73.2 | 57.9 | 64.6 |
| ANN-S1 | **81.4** | 62.4 | 70.8 |
| ANN-S2 | 78.0 | 66.3 | 71.7 |
| ANN +FrameNet | 75.1 | 59.2 | 66.2 |
| ANN-S1 +FrameNet | 80.1 | 63.6 | 70.9 |
| ANN-S2 +FrameNet | 76.8 | **67.5** | **71.9** |

Table 4: Experimental results on ACE 2005 corpus. "+FrameNet" designates the systems that are augmented by events from FrameNet.

From the results, we observe that:

1). With extra training data, *ANN* achieves significant improvements on $F_1$ measure (66.2 vs. 64.6). This result, to some extent, demonstrates the correctness of our assumption that the data sparseness problem is the reason that causes unsupervised attention mechanisms to be ineffective to ED.

2). Augmented with external data, both *ANN-S1* and *ANN-S2* achieve higher recall with a little loss of precision. This is to be expected. On the one hand, more positive training samples consequently make higher recall. On the other hand, the extra event samples are automatically extracted from FN, thus false-positive samples are inevitable to be involved, which may result in hurting the precision. Anyhow, with events from FN, our approach achieves higher $F_1$ score.

## 5 Related Work

Event detection is an increasingly hot and challenging research topic in NLP. Generally, existing approaches could roughly be divided into two groups.

The first kind of approach tackled this task under the supervision of annotated triggers and entities, but totally ignored anno-

tated arguments. The majority of existing work followed this paradigm, which includes feature-based methods and representation-based methods. Feature-based methods exploited a diverse set of strategies to convert classification clues (i.e., POS tags, dependency relations) into feature vectors (Ahn, 2006; Ji and Grishman, 2008; Patwardhan and Riloff, 2009; Gupta and Ji, 2009; Liao and Grishman, 2010; Hong et al., 2011; Liu et al., 2016b). Representation-based methods typically represent candidate event mentions by embeddings and feed them into neural networks (Chen et al., 2015; Nguyen and Grishman, 2015; Liu et al., 2016a; Nguyen and Grishman, 2016).

The second kind of approach, on the contrast, tackled event detection and argument extraction simultaneously, which is called joint approach (Riedel et al., 2009; Poon and Vanderwende, 2010; Li et al., 2013, 2014; Venugopal et al., 2014; Nguyen et al., 2016). Joint approach is proposed to capture internal and external dependencies of events, including trigger-trigger, argument-argument and trigger-argument dependencies. Theoretically, both ED and AE are expected to benefit from joint methods because triggers and arguments are jointly considered. However, in practice, existing joint methods usually only make remarkable improvements to AE, but insignificant to ED. Different from them, this work investigates the exploitation of argument information to improve the performance of ED.

## 6 Conclusions

In this work, we propose a novel approach to model argument information explicitly for ED via supervised attention mechanisms. Besides, we also investigate two strategies to construct gold attentions using the annotated arguments. To demonstrate the effectiveness of the proposed method, we systematically conduct a series of experiments on the widely used benchmark dataset ACE 2005. Moreover, we also use events from FN to augment the performance of the proposed approach. Experimental results show that our approach outperforms state-of-the-art methods, which demonstrates that the proposed approach is effective for event detection.

---

[7]http://nlp.cs.rpi.edu/software/

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
