# Peer review of "Exploiting Argument Information to Improve Event Detection via Supervised Attention Mechanisms"

_ACL 2017 — decision unknown_

[Official Review · Reviewer 1 · rating 4 · confidence 3]
soundness 5 · originality 3 · clarity 4 · substance 4 · appropriateness 5 · presentation format Poster

- Strengths:
This paper tries to use the information from arguments, which is usually
ignored yet actually quite important, to improve the performance of event
detection. The framework is clear and simple. With the help of the supervised
attention mechanism, an important method that has been used in many tasks such
as machine translation, the performance of their system outperforms the
baseline significantly.

- Weaknesses:
 The attention vector is simply the summation of two attention vectors of each
part. Maybe the attention vector could be calculated in a more appropriate
approach. For the supervised attention mechanism, two strategies are proposed.
Both of them are quite straightforward. Some more complicated strategies can
work better and can be tried.

- General Discussion:
 Although there are some places that can be improved, this paper proposed a
quite effective framework, and the performance is good. The experiment is
solid. It can be considered to be accepted.